# Mixing Performance of a Passive Micromixer Based on Multiple Baffles and Submergence Scheme

**DOI:** 10.3390/mi14051078

**Published:** 2023-05-19

**Authors:** Makhsuda Juraeva, Dong-Jin Kang

**Affiliations:** School of Mechanical Engineering, Yeungnam University, Gyoungsan 38541, Republic of Korea; mjuraeva@ynu.ac.kr

**Keywords:** degree of mixing (*DOM*), mixing energy cost (*MEC*), multiple baffles, submergence scheme, dean vortex

## Abstract

A novel passive micromixer based on multiple baffles and a submergence scheme was designed, and its mixing performance was simulated over a wide range of Reynolds numbers ranging from 0.1 to 80. The degree of mixing (*DOM*) at the outlet and the pressure drop between the inlets and outlet were used to assess the mixing performance of the present micromixer. The mixing performance of the present micromixer showed a significant enhancement over a wide range of Reynolds numbers (0.1 ≤ *Re* ≤ 80). The *DOM* was further enhanced by using a specific submergence scheme. At low Reynolds numbers (*Re* < 5), submergence scheme Sub24 produced the highest *DOM*, approximately 0.57, which was 1.38 times higher than the case with no submergence. This enhancement was due to the fluid flowing from or toward the submerged space, creating strong upward or downward flow at the cross-section. At high Reynolds numbers (*Re* > 10), the *DOM* of Sub1234 became the highest, reaching approximately 0.93 for *Re* = 20, which was 2.75 times higher than the case with no submergence. This enhancement was caused by a large vortex formed across the whole cross-section, causing vigorous mixing between the two fluids. The large vortex dragged the interface between the two fluids along the vortex perimeter, elongating the interface. The amount of submergence was optimized in terms of *DOM*, and it was independent of the number of mixing units. The optimum submergence values were 90 μm for Sub24 and *Re* = 1, 100 μm for Sub234 and *Re* = 5, and 70 μm for Sub1234 and *Re* = 20.

## 1. Introduction

Microfluidic mixing is a crucial and actively researched field that plays a vital role in enhancing fluid manipulation within various microfluidic systems. It finds applications in diverse area such as biochemistry, drug delivery, biomedical diagnostics, and chemical synthesis [1,2,3]. The design of most microfluidic systems emphasizes low reagent consumption, fast processing, low cost, and portability [3,4], making rapid and efficient mixing essential. Therefore, the overall performance of a microfluidic system depends heavily on its micromixing capability.

However, microfluidic mixing is often slow and inefficient due to molecular diffusion and slow fluid velocity in dimensions of a few hundred micrometers [2]. The fluid flow in microfluidic systems typically operates at very low Reynolds numbers, making microfluidic mixing slow and inefficient. Therefore, the development of more efficient micromixers is necessary to achieve progress in the microfluidic industry. Despite various technologies proposed to enhance microfluidic mixing, mixing enhancement remains an active area of research [2,4].

Technologies aimed at enhancing mixing in microfluidic systems are commonly classified as either active or passive. Active technologies rely on an external energy source, such as a sound field [5], magnetic field [6], electric field [7], thermal field [8,9], or pressure field [10,11]. An external energy source is primarily employed to induce fluid flow agitation and augment mixing by promoting the formation of a vortex. However, active micromixers tend to be complicated and expensive to build, limiting their wider use in microfluidic systems. In contrast, passive micromixers use geometric structures to generate chaotic fluid flow without any moving parts, making them simpler, less expensive, and more cost-effective to integrate into various microfluidic systems. Various geometric structures and modifications, such as twisting of the micromixer wall [12], a staggered herringbone design [13], blockage in the junction of the fluid stream [14], surface grooves and baffles [15,16], split-and-recombine (SAR) units [17,18], Tesla structure [19,20], stacking of the mixing units in the cross-flow direction [21], stacking in the lateral direction [22] and submergence of the mixing unit [23], have been studied to generate chaotic flow fields. However, most passive micromixers only show effective mixing in a limited range of Reynolds numbers. 

The need for a micromixer that can operate effectively in a wide range of Reynolds numbers (*Re* < 100) has arisen due to the demand for fast mixing times, in the range of milliseconds, in biological and chemical applications [24,25,26,27]. Within this range of Reynolds numbers, mixing is determined by two distinct mechanisms: molecular diffusion and convection. Consequently, micromixing can be divided into three regimes based on the dominant mixing mechanism: molecular dominance, transition, and convection dominance. Therefore, a novel design concept is needed to promote convective disturbance for flows with high Reynolds numbers and, at the same time, increase the interfacial surface of fluid layers for flows with low Reynolds numbers. 

Many researchers have attempted to enhance the mixing performance of passive micromixers by using either complex three-dimensional (3D) structures or modification of the planar geometry. While 3D micromixers may result in better mixing performance, they are more complicated and expensive to fabricate compared to planar designs [28]. Therefore, more researchers have focused on modifying planar micromixers to generate 3D flow characteristics. An example is the modified Tesla micromixer proposed by Hong et al. [29]. This micromixer design facilitates the Coanda effect, where the fluid is guided to follow the curved surface of the Tesla structure, thereby enhancing the transverse dispersion of the fluid. Tsai et al. [30] employed radial baffles in a curved microchannel to induce vortices in multiple directions. Kang [15] showed that a cyclic order arrangement of baffles along the channel wall generated vortices in the cross-flow as well as in the transverse direction. Sotowa et al. [31] enhanced the mixing performance by attaching indentations and baffles to the micromixer wall, utilizing secondary flow effects in deep micro-channel reactors. Similarly, Raza et al. [32] embedded baffles immediately following each SAR unit, resulting in enhanced mixing performance. The enhancement was noticeable in the range of Reynolds numbers from 0.1 to 80. Chung et al. [33] introduced planar baffles with side gaps to enhance the mixing performance. This modification led to significant improvements in mixing performance, demonstrated in both diffusion-dominant (*Re* < 0.1) and convection-dominant (*Re* > 40) mixing regimes. These modifications have shown significant improvements in mixing performance at low and high Reynolds numbers, but the intermediate range of Reynolds numbers (0.5 ≲Re≲20) still needs further improvement in order to develop efficient micromixers that can operate in a wide range of Reynolds numbers. 

Recently, the submergence of planar structures has been shown to be an effective technology for improving the mixing performance of 2D passive micromixers in the intermediate range of Reynolds numbers (0.1≲Re≲60). For example, Makhsuda et al. [23] showed that the submergence of planar structures generated secondary vortices in the cross-flow direction, which resulted in an additional improvement in the degree of mixing (*DOM*); 182% *DOM* increase was achieved for *Re* = 10. In addition, the submergence is easy to modify from a planar micromixer and can be fabricated using microfabrication techniques such as xurography [34] and microlithography. The xurography technique utilizes thin, pressure-sensitive, double-sided, adhesive flexible films to create a customizable submergence zone using a cutter plotter. By tailoring the film accordingly, it can be assembled with a planar structure to construct a passive micromixer in a straightforward manner. For example, Martínez-López et al. [35] demonstrated the application of xurography in the fabrication of a passive micromixer. Hsiao et al. [36] employed microlithography technology to fabricate a passive micromixer. This passive micromixer was designed with submerged winglet baffles, and polydimethylsiloxane (PDMS) material was utilized in fabrication. 

This paper proposes a novel passive micromixer that combines multiple baffles, of which some are submerged, to maximize mixing performance. The present micromixer consists of several mixing units, and each mixing unit comprises two three-quarter circles placed opposite to each other and four rectangular baffles inside. Various submergence schemes were designed and evaluated in terms of *DOM* and the associated pressure drop. *DOM* was obtained at the outlet, and the corresponding pressure drop was calculated as the pressure difference between the inlets and outlet.

This paper employed a numerical approach using commercial software to simulate the mixing performance of a proposed passive micromixer. Numerical studies have several advantages in providing easy visualization of the mixing process and flow patterns. For numerical studies, the commonly utilized software packages include ANSYS^®^ Fluent 2021 R2 [37]. For instance, Li et al. [38] employed the same software to investigate mixing performance. Additionally, COMSOL Multiphysics 5.1 (COMSOL, Inc., Burlington, MA, USA) is also widely used as another software. Rhoades et al. [39] used COMSOL to simulate the mixing performance of a grooved serpentine micro-channel. Volpe et al. [40] used the lattice Boltzmann method (LBM) to study the flow dynamics in a microfluidic device. In this paper, the mixing performance was simulated using the commercial software ANSYS^®^ Fluent 2021 R2.

## 2. Mixing Unit with Multiple Baffles

The present passive micromixer consisted of multiple mixing units, as illustrated in the schematic diagram in Figure 1. Each mixing unit comprised two mixing cells, with each mixing cell being a three-quarter circle featuring four baffles. One particular design was to use a submergence scheme that determined which baffles to submerge. For example, Figure 1 shows submergence scheme Sub24, in which the second and fourth baffles in each mixing cell were submerged. The inlet and outlet branches in the present micromixer had a rectangular cross-sectional width of 200 μm and depth of 200 μm. Inlets 1 and 2 had a length of 1000 μm each, while the outlet branch was 1200 μm in length. The two inlets were positioned facing each other, and the main micromixing process took place within the subsequent mixing units of the system. The total length of the micromixer comprising four mixing units was approximately 2.8 mm.

Figure 2 provides additional details about the mixing unit, showing several submergence schemes. Each mixing unit comprised two mixing cells, and each mixing cell was a three-quarter circle shape with four baffles, some of which were submerged based on a specific submergence scheme. For example, Figure 2a illustrates a mixing unit design with Sub24, in which the second and fourth baffles were submerged with submergence depth D_s_. The amount of submergence, represented as D_s,_ varied from 70 to 110 μm. When D_s_ was 80 μm, the height of the second and fourth baffles was 120 μm. All baffles had a thickness of 30 μm. 

Figure 2c–e depicts several examples of submergence schemes used in this paper. A submergence scheme refers to the specific configuration of which baffles are submerged. For example, Sub234 in Figure 2d indicates that the second, third, and fourth baffles were submerged. The effects of the submergence scheme on *DOM* was also studied by simulating eight different schemes, ranging from 0 to 4 submerged baffles. 

## 3. Governing Equations and Computational Procedure

The flow dynamics of the mixing process were calculated by solving the following continuity and Navier–Stokes equations:(1)$$\left(\vec{u}\cdot \nabla \right)\vec{u}=-\frac{1}{\rho }\nabla p+\nu \nabla^{2}\vec{u}$$
(2)$$\nabla \cdot \vec{u}=0$$
where $\vec{u}$, *p*, and *ν* are the velocity vector, pressure, and kinematic viscosity, respectively. The mixing process was simulated by solving the following advection–diffusion equation:(3)$$\left(\vec{u}\cdot \nabla \right)\varphi =D\nabla^{2}\varphi$$
where *D* and *φ* are the diffusion coefficient and concentration of fluid A, respectively.

The mixing process was simulated using ANSYS^®^ FLUENT 2021 R2 [37] commercial software, which is based on the finite volume method. The QUICK (quadratic upstream interpolation for convective kinematics) scheme was chosen to discretize the convective terms in the governing equations. A uniform velocity distribution was assumed at the two inlets, and the outflow condition was implimented at the outlet. All walls were modeled using a no-slip boundary condition. Specifically, at inlet 1, the mass fraction of fluid A was set to *φ* = 1, while at inlet 2, fluid B with a mass fraction of 0 was introduced.

The mixing performance of the present micromixer was assessed based on two parameters: *DOM* and mixing energy cost (*MEC*). *DOM* was calculated using the following formula:(4)$$DOM=1-\frac{1}{\xi }\sqrt{\overset{n}{\underset{i=1}{\sum }}\frac{{\left(\varphi_{i}-\xi \right)}^{2}}{n}},$$
where *φ_i_* and *n* are the mass concentration of fluid A of *i*th cell and the total number of cells, respectively. In this case, *ξ* was set to 0.5, representing the complete mixing of two fluids. *DOM* = 1 indicates that complete mixing has been achieved and the fluids are homogenized. Conversely, *DOM* = 0 denotes complete separation of fluids, with no mixing occurring. *MEC* was used to evaluate the effectiveness of the present micromixer and was calculated in the following form [41,42]:(5)MEC=Δpρumean2DOM×100,
where $u_{mean}$ is the average velocity in the micromixer, and Δp is the pressure drop between the inlets and the outlet. For a given value of *DOM*, a smaller *MEC* indicated more efficient mixing as it required less pressure load (energy input) to achieve the desired level of mixing.

The properties of the fluid flowing into both inlets, including the density, diffusion coefficient, and viscosity, were assumed to be identical to those of water, with values of *ρ* = 998 kg/m^3^, *D* = 1.0 × 10^−10^ m^2^ s^−1^, and *ν =* 1.0 × 10^−6^ m^2^ s^−1^, respectively. The value of the diffusion coefficient was the same as that used in other studies [23,28]. The Reynolds number was defined as $Re=\frac{\rho U_{mean}d_{h}}{\mu }$, where $\rho ,\;\;U_{mean},\;\;d_{h},\;\;\mathrm{and}\;\mu$ indicate the density, the mean velocity at the outlet, the hydraulic diameter of the outlet channel, and the dynamic viscosity of the fluid, respectively The Schmidt (Sc) number was 10^4^.

## 4. Validation of the Numerical Study

In general, high Sc (Schmidt number) simulations may suffer from numerical diffusion, which can deteriorate the accuracy of numerical solutions. Nevertheless, this computational issue is not often extensively addressed in many papers. To obtain a more quantitatively rigorous numerical solution, two possible remedies can be considered. One option is to utilize a particle-based simulation method such as the Monte Carlo method [43]. This approach can provide more accurate results by explicitly modeling the behavior of individual particles, thereby reducing the impact of numerical diffusion. Alternatively, in grid-based methods, another remedy involves decreasing the cell Peclet number. The cell Peclet number is defined as $Pe_{c}=\frac{U_{cell}l_{cell}}{D}$, where $U_{cell}$ represents the local flow velocity and $l_{cell}$ indicates the size of the individual cells in the grid. By reducing the cell Peclet number, the influence of numerical diffusion can be mitigated, leading to more accurate numerical solutions. For example, Bayareh [44] suggested keeping the cell Peclet number $Pe_{c}\leq 2$ to obtain a numerical solution with negligible numerical diffusion effects. However, these options were too computationally expensive to adopt in studies such as that described in this paper. As a more practical approach, most numerical studies carry out a grid independence test [45].

In a previous study [23], the present numerical approach was thoroughly validated through simulations of a passive micromixer conducted by Tsai et al. [30]. Figure 3 displays a schematic diagram of the simulated micromixer, which had a rectangular cross-sectional width of 45 μm and depth of 130 μm for both inlets. Additional information can be found in [23,30].

Figure 4 illustrates a quantitative comparison between the numerical results and the corresponding experimental data by Tsai et al. [30]. The comparison is performed within a Reynolds number range spanning 1 to 81. Here, *DOM_T_* was the degree of mixing used by Tsai et al. [30], defined in the following way:(6)$$DOM_{T}=1-\frac{\sigma_{D}}{\sigma_{D,o}},$$
and
(7)$$\sigma =\sqrt{\frac{1}{n}\sum_{i=1}^{n}{\left(\varphi_{i}-\varphi_{ave}\right)}^{2}},$$
where $\sigma_{D}$ is the standard deviation of φ on a cross-section normal to the flow, $\sigma_{D,o}$ is calculated at the inlet, and $\varphi_{ave}$ is the average value of φ at a cross-section under investigation.

Despite some discrepancies between the numerical solution and experimental data, similar behavior was observed with respect to the Reynolds number. The discrepancy was below 4% and it decreased as the Reynolds number and Peclet number increased. The observed discrepancy could be attributed to several factors, including numerical diffusion and experimental uncertainty. The numerical results were also compared in terms of mixing image. Figure 5 presents a comparison between the numerical concentration contours on the horizontal mid-plane and the experimental confocal images at Reynolds numbers of 1, 9, and 81.

To ensure the accuracy of the present numerical solutions in this study, preliminary simulations were conducted to determine an appropriate edge size for the micromixer under investigation. The edge size of the cells was varied from 4 to 6 μm for a micromixer having five mixing units, resulting in a corresponding number of cells ranging from 2.14 × 10^6^ to 7.36 × 10^6^. Figure 6 displays an enlarged grid in a mixing cell. The simulation was performed for, Sub1234 with *Re* = 1 and D_s_ = 90 μm. According to Okuducu et al. [46], the type of cells affects the accuracy of numerical solutions. Among different cell types such as tetrahedral, prism, and hexahedral cells, structured hexahedral cells were predominantly used in this study due to their capability to provide more reliable numerical solutions. This can be observed in Figure 6. The number of prism cells was minimized, as demonstrated by the red circle in the figure.

The grid convergence index (*GCI*) was calculated using the simulation results to quantify the uncertainty associated with the numerical solution [47,48]. The *GCI* was calculated using the Richardson extrapolation methodology and is expressed as:(6)$$GCI=F_{s}\frac{\left|\varepsilon \right|}{r^{p}-1},$$
(7)$$\varepsilon =\frac{f_{coarse}-f_{fine}}{f_{fine}},$$
where *F_s_* and *p* are the safety factor and the accuracy order of the numerical method, respectively; *r* is the grid refinement ratio; and *f_coarse_* and *f_fine_* are the numerical solutions obtained with coarse and fine grids, respectively. In this case, *F_s_* was specified at 1.25 as suggested by Roache [47]. For edge sizes of 4, 5, and 6 μm, the corresponding numbers of cells were 2.14 × 10^6^, 3.76 × 10^6^, and 7.35 × 10^6^, respectively. The *GCI* of the computed *DOM* was reduced from 5.9% to 2.9%. Considering computational accuracy and cost, the edge size of 5 μm was chosen to obtain the present numerical solutions.

## 5. Results and Discussion

The mixing performance of the present micromixer was first simulated to investigate the effects of the submergence scheme in the Reynolds number range of 0.1≤Re≤80, for which the corresponding Peclet number range was $10^{3}\leq Pe\leq 8\times 10^{4}$. The number of submerged baffles in each mixing cell was varied from 0 to 4, based on a specific submergence scheme. A value of 0 denoted the absence of any submerged baffles, while a value of 4 indicated that all baffles were submerged within the micromixer. The velocity at the two inlets was uniformly set within a range of 0.25 mm/s to 0.2 m/s, which corresponded to a volume flow rate ranging from 1.2 to 964.6 μL/min. The mixing performance was assessed based on the *DOM* at the outlet and the corresponding pressure drop.

Passive micromixers achieve mixing enhancement primarily through a significant increase in pressure load, which should be considered when evaluating their performance. Figure 7 shows a mixing performance map as a function of Reynolds number, where four typical planar micromixers simulated by Raza et al. [32] for the same physical and boundary conditions as those used in this paper were compared directly. The present numerical results were obtained with the submergence depth of D_s_ = 90 μm. The present micromixer exhibited significant *DOM* enhancement in the low and intermediate range of Reynolds numbers (Re≤ 20). Conversely, it required the lowest pressure load over a wide range of Reynolds numbers (0.1≤Re≤80), except that of SAR with dislocation [38], which exhibited the worst *DOM* among the passive micromixers discussed in this paper. For example, the required pressure load of the present micromixer was merely 23% of that of modified Tesla micromixer [19], which showed a comparable *DOM* for the Reynolds number (Re=80). This result suggested that the present micromixer offered highly effective mixing performance over a broad range of Reynolds numbers (0.1≤Re≤80).

Figure 8 shows the effects of the number of submerged baffles on the *DOM* at the outlet. While submerged baffles generally resulted in enhanced *DOM*, the magnitude of this enhancement was highly dependent on the number of submerged baffles. For example, the *DOM* of Sub24 was approximately 0.57 for *Re* = 1, corresponding to 1.38 times the value of the case with no submergence. On the other hand, the *DOM* of Sub1234 was approximately 0.93 for *Re* = 20, which corresponded to 2.75 times the value of the case with no submergence. At low Reynolds numbers (*Re* < 5), the maximum *DOM* was obtained when two baffles were submerged. In contrast, the highest *DOM* was achieved when all baffles were submerged at high Reynolds numbers (Re>10). In the intermediate range of Reynolds numbers (5≤Re < 10), the submergence of three baffles (Sub234 in Figure 8) exhibited the highest *DOM*. This result suggested that the submergence of baffles became increasing effective as the Reynolds number rose. 

Figure 9 depicts the effects of the different submergence schemes when two baffles were submerged. The results indicated that Sub24 exhibited the highest *DOM* in the Reynolds number range of Re≤ 10, whereas Sub14 performed best for Reynolds numbers greater than 10. Noticeably, the *DOM* of Sub13 was similar to that of Sub24 in the Reynolds number range of Re≤ 1. Therefore, alternating submergence scheme exhibits the best performance in terms of *DOM*, particularly in the low Reynolds number range of Re≤ 1.

The efficiency of the submerged baffles was also analyzed in terms of *MEC*, as shown in Figure 10. The *MEC* distribution was compared as a function of Reynolds number. All of the submergence schemes showed improved *MEC* distribution compared to the case with no submergence, where a smaller *MEC* signified a reduced pressure load needed to achieve a given *DOM*. Sub24 exhibited the smallest *MEC* in the low Reynolds number range, whereas Sub1234 resulted in the smallest value in the high Reynolds number range. For example, the *MEC* of Sub24 at *Re* = 1 was approximately 30.27, which was a 59% reduction from the value of the no submergence case. The *MEC* of Sub1234 at *Re* = 20 was 1.05, which was roughly one-fifth of the value of the no submergence case. This reduction suggested that the submergence of baffles required a lesser pressure load to achieve a specific value of *DOM*. 

As the height of submerged baffles is an important design parameter, its effects on mixing performance were also analyzed in detail. Figure 11 shows that the enhancement of *DOM* was optimized based on the amount of submergence for a specific submergence scheme. For example, the optimum submergence values were 90 μm for Sub24 and *Re* = 1, 100 μm for Sub234 and *Re* = 5, and 70 μm for Sub1234 and *Re* = 20. Notably, the optimum submergence value was independent of the number of mixing units. Another interesting finding was that the *DOM* enhancement rate with the number of mixing units was steeper when a submergence scheme of baffles was used, especially in the low and intermediate ranges of Reynolds numbers. For example, the slope of the broken dotted line was 2.9 times steeper than that of the dotted line in Figure 11a for Sub24 and *Re* = 1. Similarly, the slope of the submergence case was about 4 times steeper than that of the no submergence case in Figure 11b for Sub234 and *Re* = 5.

The study further investigated the mixing enhancement at three Reynolds numbers of *Re* = 1, 5, and 20, where the three different submergence schemes exhibited the best mixing performance: Sub24, Sub234, and Sub 1234. Figure 12 compares the concentration contours on the xy-plane at z = 100 μm for three Reynolds numbers of *Re* = 1, 5, and 20. The plane located at z = 100 μm corresponded to the mid-depth plane of the micromixer under investigation. For no submergence cases, the mixing due to chaotic advection with an increase in the Reynolds number seemed limited. Mixing was promoted only in the region between two consecutive baffles. Therefore, the flow along the outer circular wall remained unmixed; Figure 12a shows that the mixed and unmixed zones were clearly separated. In contrast, for submergence cases, the mixing due to chaotic advection was noticeably apparent as the Reynolds number increased. The mixing enhancement was obvious in the both the radial direction and along the micromixer, even for the Reynolds number of *Re* = 1. The mixing between two consecutive baffles was more rigorous.

Figure 13 explains how submergence scheme Sub24 changed the flow pattern in the first mixing cell. When submergence scheme Sub24 was applied, the fluid between the second and third baffles in the submerged space was directed to flow outward; the plane at z = 50 μm was located at the mid-depth of submergence. In contrast, for the no submergence case, the fluid between the second and third baffles was almost trapped inside.

Figure 14 illustrates the effects of the submergence scheme on the concentration distribution for three Reynolds numbers of *Re* = 1, 5, and 20; the cross-section is the yz-plane at A, B, C, and D in the figure. For the no submergence case, the mixing in the y and z directions was quite limited for *Re* = 1 and 5. When the Reynolds number was increased to *Re* = 20, a pair of Dean vortexes formed at the cross-section, causing noticeable mixing at the cross-section. On the contrary, for submerged cases, the mixing at the cross-section was clearly observed even in the first mixing unit for the Reynolds number of *Re* = 1. The interface on the left-hand side for *Re* = 1 in Figure 14b was wider than that on the right-hand side. This was caused by the fluid flow change explained in Figure 13. This difference of concentration became more obvious in the downstream direction. A similar pattern of mixing was observed for the Reynolds number of *Re* = 5. When the Reynolds number was increased to *Re* = 20, submergence scheme Sub1234 generated a large vortex across the entire cross-section instead of a Dean vortex. This resulted in more vigorous mixing between the two fluids. Due to the action of the large vortex, the interface at the cross-section was more elongated and dragged along the large vortex, as depicted in Figure 14b. 

Figure 15 demonstrates the effects of the submergence scheme on the vortex flow pattern within the cross section located at the position D in Figure 14. For no submergence case, a pair of Dean vortices was formed on the upper side of the cross section. However, for the submergence scheme Sub1234, the left vortex becomes dominant, spanning the entire cross section. On the other hand, the Sub24 and Sub234 schemes disrupted one vortex, resulting in a flow either upward or downward. Among the various flow patterns observed on the cross section, the large vortex exhibited the best mixing performance, as described previously.

Figure 16 illustrates how the large vortex improved mixing along the micromixer. For the no submergence case, the streamlines starting from both inlets travelled calmly to the outlet of the micromixer. In contrast, the streamlines for submergence scheme Sub 1234 travelled widely in the z-direction due to the large vortex motion at the cross-section. This flow pattern was the main mechanism of the mixing enhancement. When submergence scheme Sub1234 was used, the *DOM* at the outlet was 0.93, representing a 258% enhancement compared to the value of the no submergence case, for which the *DOM* was only 0.25. Meanwhile, the corresponding pressure drop was reduced from 2280 (pa) to 1519.5 (pa).

## 6. Conclusions

This paper proposed a passive micromixer that utilized multiple baffles and a submergence scheme. The proposed micromixer consisted of multiple mixing units, each of which had two mixing cells with four baffles arranged in a specific submergence scheme determining which baffles were submerged. Each mixing cell was in the shape of a three-quarter circle. To evaluate the mixing performance of the present micromixer, numerical simulations were conducted using ANSYS^®^ Fluent 2021 R2. The simulations encompassed a range of Reynolds numbers from 0.1 to 80, and three different configurations were examined with varying numbers of mixing units from 3 to 5. The assessment of mixing performance was conducted by evaluating the *DOM* at the outlet. The corresponding pressure drop was also considered as a metric to assess the mixing efficiency.

The present micromixer demonstrated a significant enhancement of *DOM* over a wide range of Reynolds numbers (Re≤ 20) compared to typical planar passive micromixers, while requiring a low pressure load. The optimum submergence scheme for achieving the highest *DOM* depended on the Reynolds number. At low Reynolds numbers (*Re* < 5), the highest *DOM* was achieved when two baffles were submerged in an alternating order. In contrast, at high Reynolds numbers (Re>10 ), the highest *DOM* was observed when all four baffles were submerged. For Reynolds numbers in the intermediate range (5≤Re≤ 10), the best submergence scheme involved the submergence of three baffles.

The mixing performance of the present micromixer was further enhanced by controlling the submergence depth. The submergence depth was optimized based on the *DOM* at the outlet, with optimum submergence values of 90 μm for Sub24, 100 μm for Sub234, and 70 μm for Sub1234. However, the optimum value of submergence was almost independent of the number of mixing units and Reynolds number. The *DOM* enhancement rate with the number of mixing units became steeper when using a submergence scheme at low and intermediate ranges of Reynolds numbers.

The present micromixer demonstrated enhanced mixing performance over a wide range of Reynolds numbers. This enhancement was mainly achieved through the use of a specific submergence scheme tailored to the Reynolds number. This submergence scheme was easy to fabricate by lowering the height of the baffles inside the mixing cells.

## Figures and Tables

**Figure 1 micromachines-14-01078-f001:**
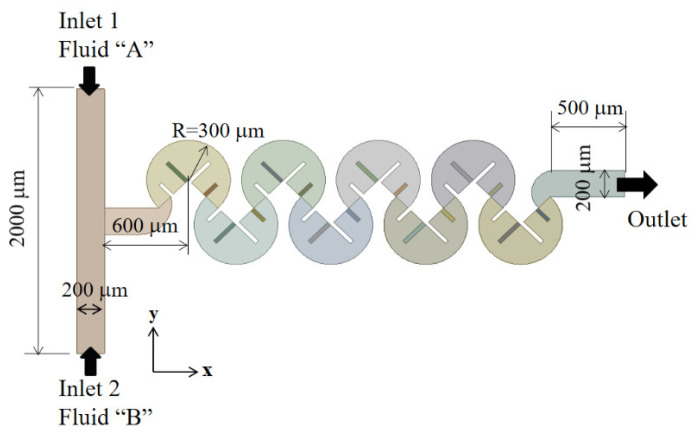
Schematic of the present micromixer.

**Figure 2 micromachines-14-01078-f002:**
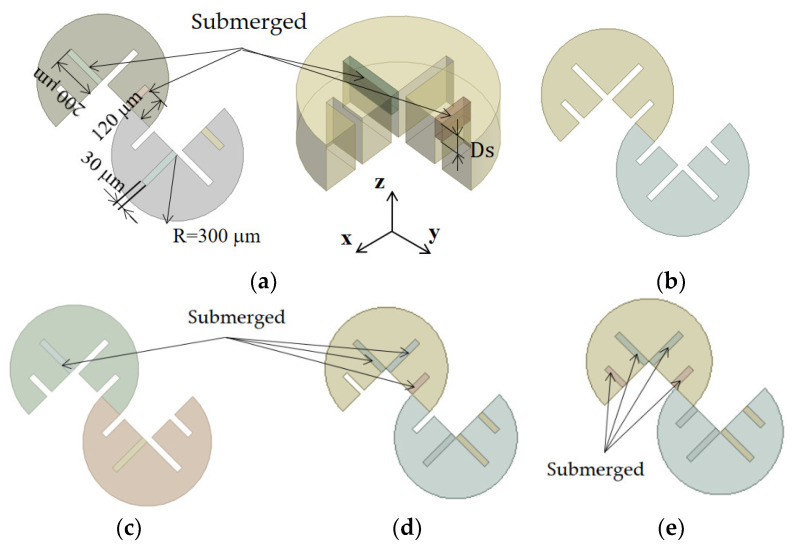
Submergence schemes: (**a**) Sub24, (**b**) no submergence, (**c**) Sub2, (**d**) Sub234, and (**e**) Sub1234.

**Figure 3 micromachines-14-01078-f003:**
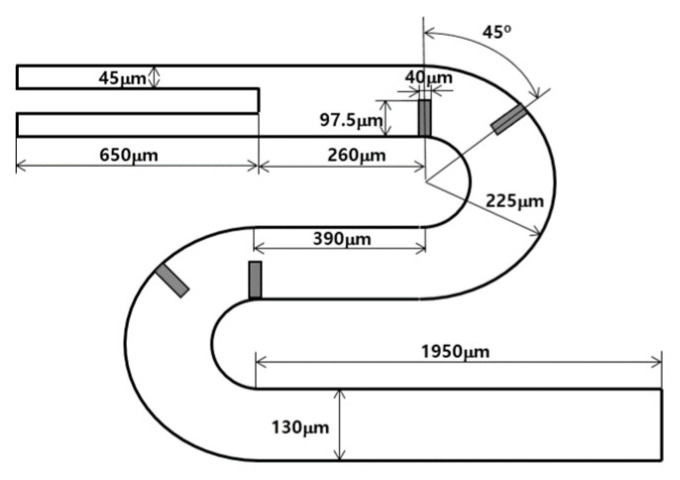
Schematic of the micromixer tested by Tsai et al. [30].

**Figure 4 micromachines-14-01078-f004:**
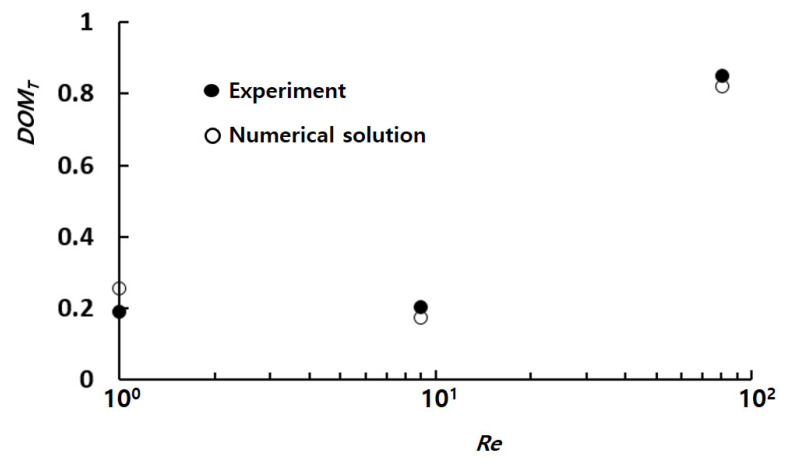
Comparison of present numerical solutions with corresponding experimental data [30].

**Figure 5 micromachines-14-01078-f005:**
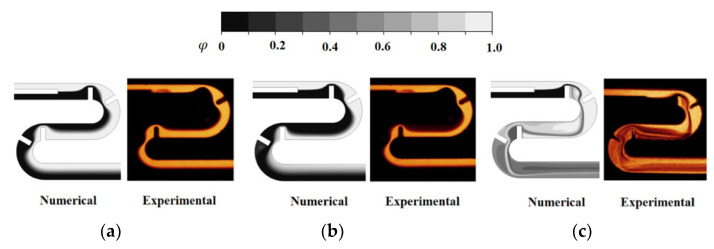
Comparison of numerical concentration contours with experimental images: (**a**) *Re* = 1, (**b**) *Re* = 9, and (**c**) *Re* = 81.

**Figure 6 micromachines-14-01078-f006:**
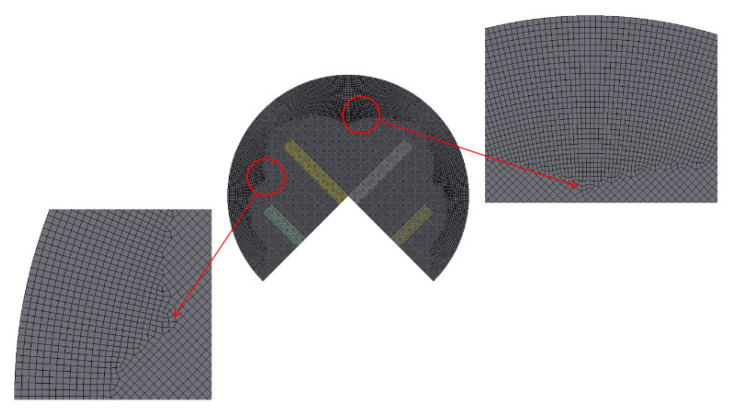
Example of grid in a mixing cell.

**Figure 7 micromachines-14-01078-f007:**
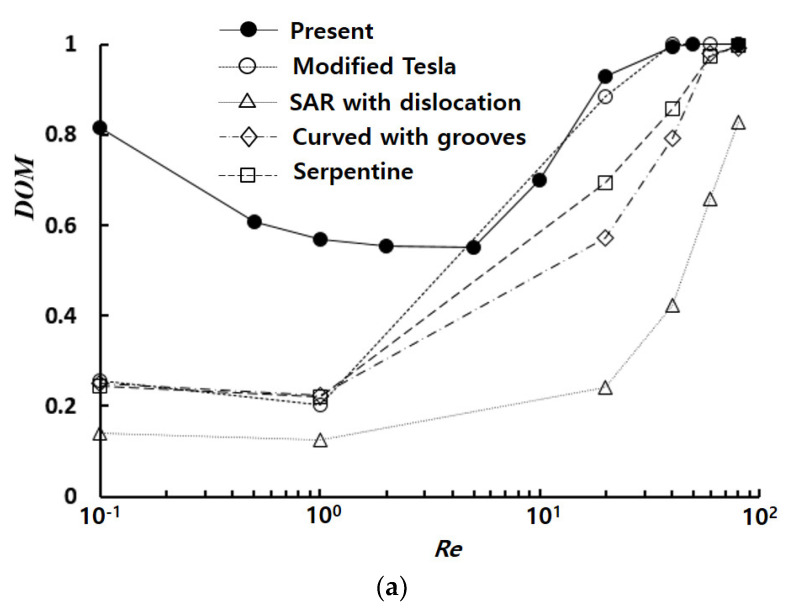
Comparison of present mixing performance with typical passive micromixers: (**a**) *DOM*, and (**b**) pressure load. Modified Tesla micromixer by Hossain et al. [19]; SAR with dislocation by Li et al. [38]; Curved with grooves by Alan et al. [49]; Serpentine by Hossain et al. [50].

**Figure 8 micromachines-14-01078-f008:**
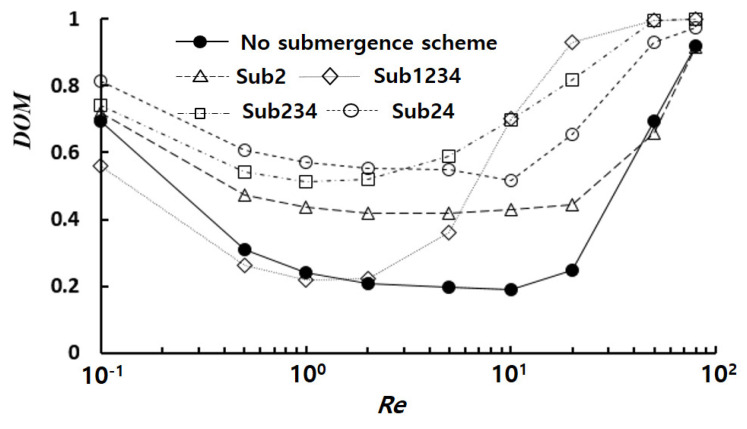
Effects of the number of submerged baffles on the *DOM*.

**Figure 9 micromachines-14-01078-f009:**
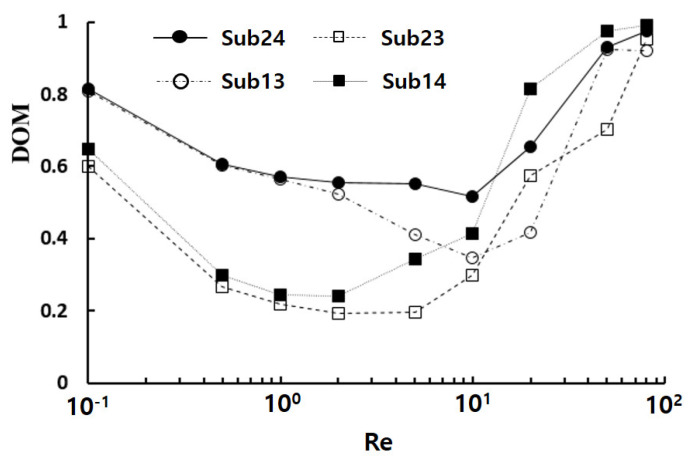
Effects of the submergence scheme on the *DOM*.

**Figure 10 micromachines-14-01078-f010:**
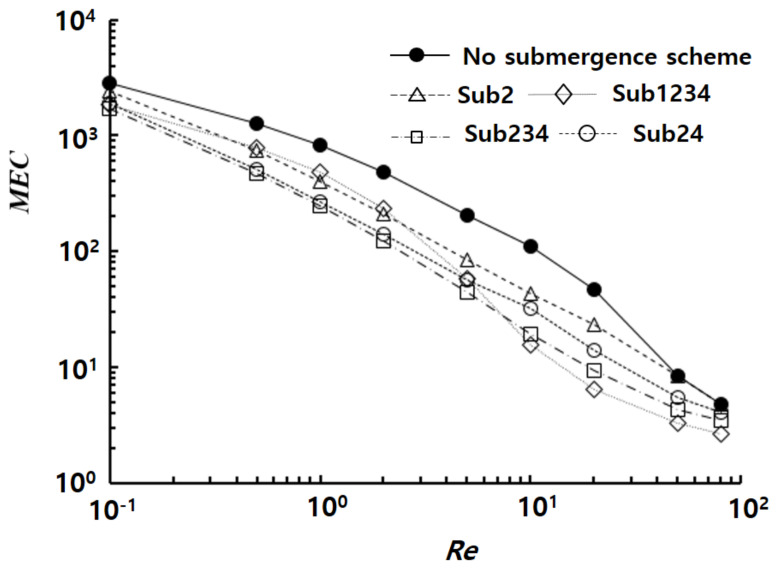
Effects of the submergence scheme on the *MEC*.

**Figure 11 micromachines-14-01078-f011:**
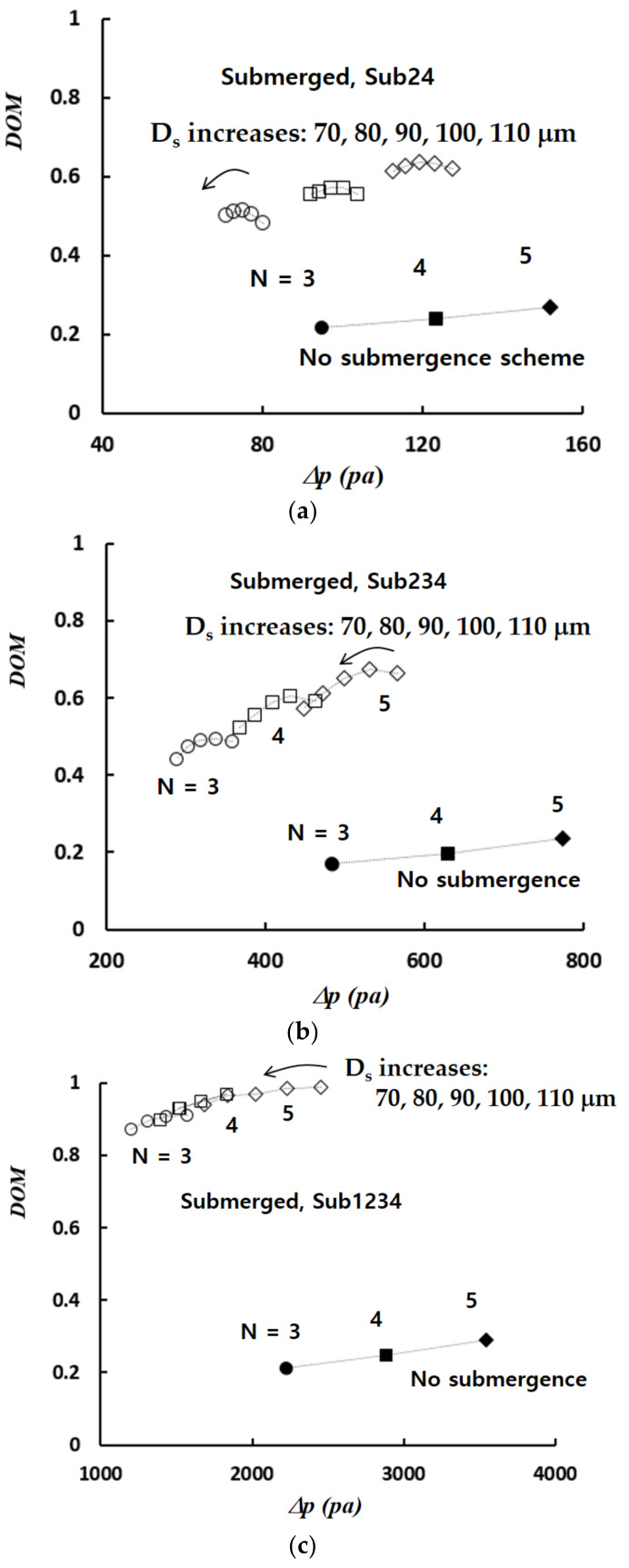
Mixing performance map as a function of the number of mixing units and submergence scheme: (**a**) *Re* = 0.1, (**b**) *Re* = 5, and (**c**) *Re* = 20.

**Figure 12 micromachines-14-01078-f012:**
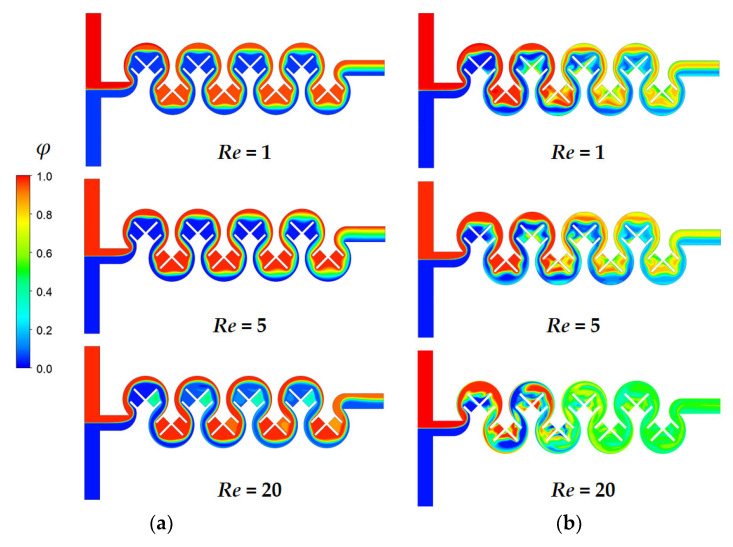
Concentration distribution on the xy-plane at z = 100 μm: (**a**) no submergence and (**b**) submergence scheme with *D_s_* = 100 μm.

**Figure 13 micromachines-14-01078-f013:**
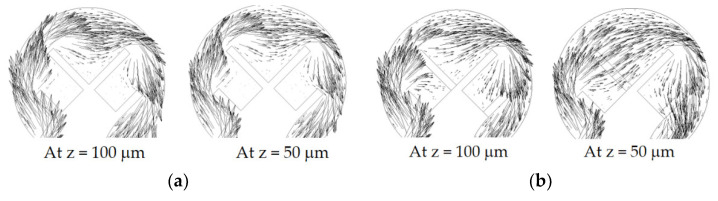
Velocity vector field in the first mixing cell for *Re* = 1: (**a**) no submergence, and (**b**) submergence scheme Sub 24 with *D_s_* = 100 μm.

**Figure 14 micromachines-14-01078-f014:**
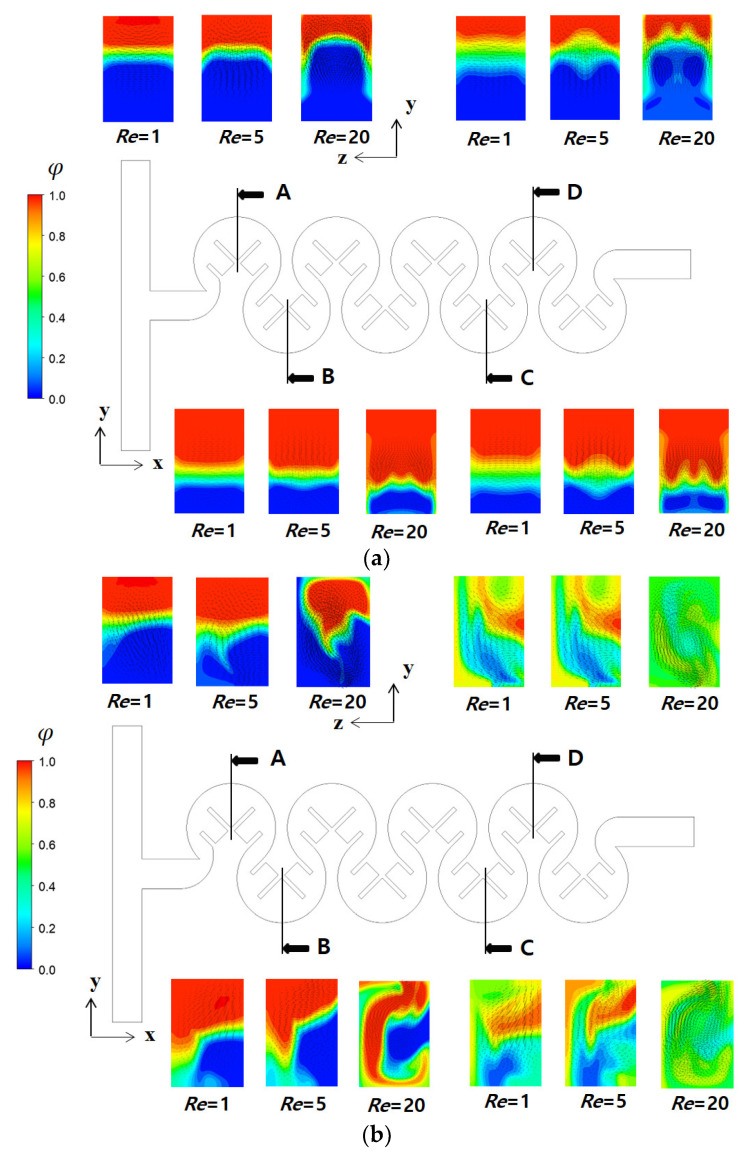
Comparison of concentration distribution at the cross-section: (**a**) no submergence, and (**b**) submergence scheme with *D_s_* = 100 μm.

**Figure 15 micromachines-14-01078-f015:**
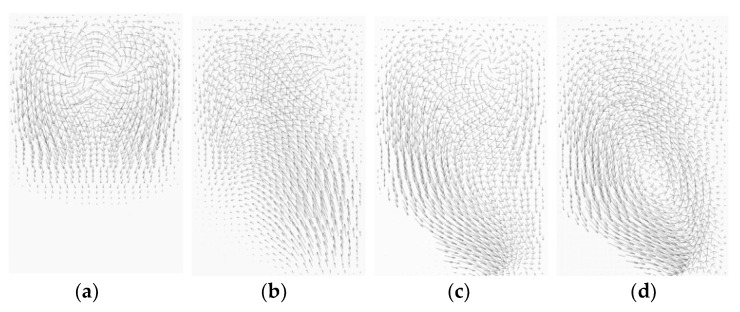
Effects of submergence scheme on the vortex flow pattern at the cross-section for *Re* = 20: (**a**) no submergence, (**b**) Sub24, (**c**) Sub234, and (**d**) Sub1234 with *D_s_* = 100 μm.

**Figure 16 micromachines-14-01078-f016:**
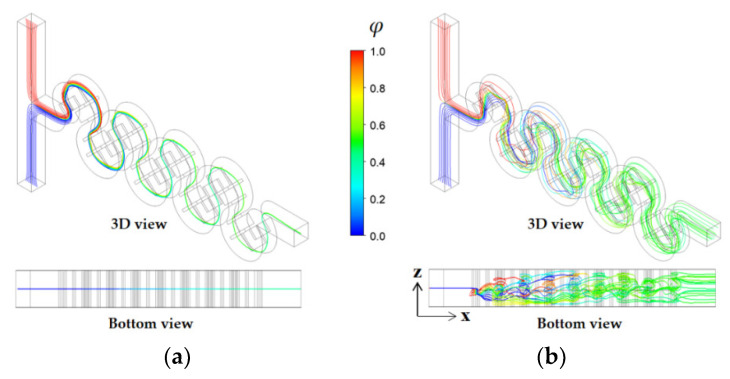
Comparison of pathlines starting from the inlets for *Re* = 20: (**a**) no submergence and (**b**) submergence scheme Sub1234 with *D_s_* = 100 μm.

## Data Availability

Not applicable.

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
