# Peer review of "Mixing Performance of a Passive Micromixer Based on Multiple Baffles and Submergence Scheme"

_micromachines, 2023, doi:10.3390/mi14051078_

Round 1
Reviewer 1 Report
The manuscript should address the following comments:
1. From the understanding of the reviewer, in equation (7), symbol "psi" represents concentration of sample, which is not consistent with the symbol used in equation (3)
2. In figure 5, it is not clear what is the scale of the contour
3. Figure 7, 8, 9, 10, 11 must be improved to show the data and trend clearer
4. In figure 12, 14-19, the legend should be given to show the range of the colored contour
The quality of English language is fine
Reviewer 2 Report
This paper proposes a novel design for a passive micromixer that utilizes multiple baffles and a submergence scheme. The simulations considered Reynolds numbers from 0.1 to 80 and three different numbers of mixing units ranging from 3 to 5. The mixing performance was assessed based on the DOM at the outlet and the required pressure load between the inlets and the outlet. The simulation results show that the present micromixer demonstrates a significant enhancement of DOM over a wide range of Reynolds numbers (Re ≤ 20) compared to typical planar passive micromixers, while requiring a low pressure load. This is a very interesting work, but the following issues need further clarification by the authors.
1. Line91 mentions the method of processing the structure, whether there is a manufactured physical object, or provides other proof that the structure can be actually processed, which is important for the practical application of the structure.
2. More understandable instructions are needed for the baffle immersion scheme. The submerged area means that there is a solid baffle present or that the area is for fluid to flow through? For the sub24 scheme, do the first and third baffles vertically occupy the entire channel cross-section? Figure 2 should illustrate the information more easily and understandably.
3. Authors should consider adding further descriptions of the DOM and MEC, including the range of their correct calculation results, and what larger or smaller DOMs and MECs mean.
4. Please provide references for the physical properties parameters in Line184.
5. Please check if the serial number of the figure in Line213 "Additional information can be found in Figure 4." is correct.
6. Line261, why the number of corresponding nodes increases first and then decreases as the edge size increases, what is an acceptable GCI, and why the final size is chosen to be 5μm.
7. The structural parameters used for comparison of present structure in Figure 7 should be described.
8. Please further explain the conclusion of Line313.
9. Which results are used to make the conclusions at Line350? The range of Ds highlighted at Line139 is 40μm-80μm, do the statements at Line250 and Line350 conflict with this? Another question is, how does the optimal submergence depth vary with Reynolds number, in addition to being influenced by the submergence scheme?
10. The horizontal coordinate in Figure 11 shows the variation in Reynolds number, which does not match the description in the figure notes and the range of 0.1-80 discussed in the paper, please explain again what the increase in the horizontal coordinate represents. Also, what is the range of variation of Ds in Fig. 11?
11. Please align figure (a) and (b) in Figure 13.
12. The quantitative velocity field distribution in the channel is one of the issues that should be investigated, which is feasible for numerical simulations. The authors should consider adding velocity field studies in the channel. For example, a comparison of velocity magnitudes (quantitative) in the downstream direction or cross-section of the channel for different submergence schemes, or which part of the channel velocity distribution is changed by structural variations, which would be beneficial to increase the completeness of the study.
Minor editing of English language required
Reviewer 3 Report
The manuscript entitled Mixing Performance of a Novel Passive Micromixer based on Multiple Baffles is numerical study of micro-mixing. I commented as follows;
1.Reynolds number should be shown in power index notation.
2.The author should discuss Peclet number.
3.The many symbols and letters were used. The author should summarize them as a nomenclature.
Reviewer 4 Report
In the present study, a novel passive micromixer based on submerged baffles (a micromixer design concept proposed by authors in a previous paper [23]) is proposed, and analyzed numerically using commercial CFD package ANSYS – FLUENT 2021 R2. A relative evaluation of different submerged designs is presented in terms of mixing performance and pressure drop.
The authors have conducted a systematic parametric to determine the most successful submerged baffle configuration. Some comments to the authors:
1. The study is extension of the previous design concept [23], and therefore, parametrization seems more appropriate than novelty.
2. The authors already compared the prediction accuracy of numerical methods used in the present work in their previous research [23]. Section 4 – validation of numerical study needs to be revised to remove redundant information with proper citation to already published results from [23].
3. In Figure 7 (a), the legends need to be adjusted to bottom – left OR bottom – right for clarity to the readers.
4. Figure 17, bottom z = 50 μm is missing.
5. It would be better to combine the concentration contour plots (Figs. 12, 15 and 17) with different submergence schemes at one particular location, either z = 50 μm or 100 μm (I don’t see much difference in contours at two locations, except the contour levels) for better understanding the effect of submergence schemes on mixing.
6. Similar to (5), Figs. 14, 16 and 18 can be combined as single figure.
7. Provide contour bars, either horizontal or vertical for all concentration contour plots.
8. The conclusion is superfluous and must be revised for brevity.
I urge the authors to re-write the paper for effective presentation and Readability by doing a moderate revision of English language.
